# Mitochondrial Volume Regulation and Swelling Mechanisms in Cardiomyocytes

**DOI:** 10.3390/antiox12081517

**Published:** 2023-07-28

**Authors:** Xavier R. Chapa-Dubocq, Keishla M. Rodríguez-Graciani, Nelson Escobales, Sabzali Javadov

**Affiliations:** Department of Physiology, University of Puerto Rico School of Medicine, San Juan, PR 00936-5067, USA; xavier.chapa@upr.edu (X.R.C.-D.); keishla.rodriguez20@upr.edu (K.M.R.-G.); nelson.escobales@upr.edu (N.E.)

**Keywords:** mitochondria, ions, permeability transition pore, heart, ischemia-reperfusion

## Abstract

Mitochondrion, known as the “powerhouse” of the cell, regulates ion homeostasis, redox state, cell proliferation and differentiation, and lipid synthesis. The inner mitochondrial membrane (IMM) controls mitochondrial metabolism and function. It possesses high levels of proteins that account for ~70% of the membrane mass and are involved in the electron transport chain, oxidative phosphorylation, energy transfer, and ion transport, among others. The mitochondrial matrix volume plays a crucial role in IMM remodeling. Several ion transport mechanisms, particularly K^+^ and Ca^2+^, regulate matrix volume. Small increases in matrix volume through IMM alterations can activate mitochondrial respiration, whereas excessive swelling can impair the IMM topology and initiates mitochondria-mediated cell death. The opening of mitochondrial permeability transition pores, the well-characterized phenomenon with unknown molecular identity, in low- and high-conductance modes are involved in physiological and pathological increases of matrix volume. Despite extensive studies, the precise mechanisms underlying changes in matrix volume and IMM structural remodeling in response to energy and oxidative stressors remain unknown. This review summarizes and discusses previous studies on the mechanisms involved in regulating mitochondrial matrix volume, IMM remodeling, and the crosstalk between these processes.

## 1. Introduction

Mitochondria are intracellular organelles comprised of two membranes: the inner mitochondrial membrane (IMM) and the outer mitochondrial membrane (OMM). These organelles are responsible for ~90% of the cellular ATP production. Additionally, mitochondria maintain cellular ion homeostasis and redox balance, participate in lipid metabolism, generate proteins and DNA precursors, and produce reactive oxygen species (ROS). In the heart, a high-energy consuming organ, mitochondria play a critical role in maintaining myofibrillar contractility, cellular ion homeostasis, and multiple anabolic activities [1]. For these reasons, various mechanisms operate to control cardiac oxidative and energy stress. Cardiac muscle cells (cardiomyocytes) contain up to 5000 mitochondria [2]. They are divided into three types based on their location within subcellular compartments, including the intermyofibrillar, subsarcolemmal, and perinuclear mitochondria [3]. Subcellular distribution allows mitochondria to supply ATP to myofibrils and control Ca^2+^ flux, ROS, and the NAD^+^/NADH ratio, thereby coupling redox equilibrium to energy demand [4]. Furthermore, mitochondria actively participate in several cell death mechanisms, including apoptosis, necroptosis, pyroptosis, ferroptosis, and autophagy-induced cell death [5].

Mitochondrial Ca^2+^ regulates energy metabolism under physiological conditions, but when Ca^2+^ levels are markedly increased, it leads to mitochondrial fragmentation and mitochondria-mediated cell death [6]. Mitochondrial Ca^2+^ overload, associated with increased ROS generation, ATP depletion, and high P_i_ levels, occurs during reperfusion after myocardial infarction and ischemia. Indeed, these perturbances lead to ischemia-reperfusion (IR) injury of the heart by opening non-selective, high-conductance mitochondrial permeability transition pores (mPTP) in the IMM. mPTP opening allows solutes with molecular mass up to 1.5 kDa to pass through the IMM, jeopardizing energy metabolism and structural integrity of mitochondria through osmotic volume swelling. The mitochondrial chaperone protein peptidyl-prolyl cis-trans isomerase cyclophilin D (CypD) regulates mPTP opening, although the molecular identity of the pore remains unclear. The therapeutic significance of mPTP as a modulator of mitochondrial matrix volume has gained significant interest over the years. This interest stems from its therapeutic potential to ameliorate ischemic cardiac disease and enhance recovery after myocardial infarction.

Mitochondrial ion transport mechanisms also play a crucial role in regulating mitochondrial bioenergetics through changes in matrix Ca^2+^ levels that, in turn, affect mitochondrial respiration and ATP generation under physiological conditions [7]. Together with IMM remodeling, these processes assure a normal bioenergetic status critical for responding to stressors and maintaining cellular homeostasis. It is likely that disruption of the interplay between matrix volume changes and IMM remodeling by oxidative/energy stress during IR results in mitochondrial and cardiac dysfunction. This review explores the possible mechanisms by which mitochondrial volume and the structural integrity of the IMM are regulated. The information presented could be important for developing new approaches to mitigate cell death secondary to mitochondrial swelling.

## 2. Mitochondrial Membrane Structure and Organization

### 2.1. Mitochondrial Membrane Composition

The complex construction of the mitochondrial membranes reflects their functional specialization. Both the OMM and IMM possess transport mechanisms for proteins and metabolites that regulate the structural organization of mitochondria. The OMM is primarily responsible for allocating proteins for specific functions of mitochondria (i.e., protein translocation, mitochondrial quality control, among others) through the regulation of mitophagy and mitochondrial dynamics [8]. In contrast to the OMM, the IMM has a more complex structural organization. It is divided into the inner boundary membrane, a part of the IMM adjacent to the OMM, and the cristae membrane, which forms invaginations projecting into the mitochondrial matrix. Both subdomains communicate through narrow and tubular membrane segments called cristae junctions (CJs), which attach the inner boundary and cristae membranes [9] (Figure 1). Due to heterogeneity in protein composition, membrane structural dynamics, and phospholipid biogenesis, these two membrane domains are morphologically and functionally distinct [9]. The inner boundary membrane, which runs parallel to the OMM, is considered a secondary envelope structure containing protein import machinery close to those present in the OMM, thereby facilitating protein transport into the matrix. The cristae membrane possesses a distinctive fold-like structure that significantly expands its surface area. This topographical feature enables the efficient organization of numerous protein complexes crucial for energy generation through the electron transport chain (ETC) and oxidative phosphorylation (OXPHOS) [10].

Mitochondria are a major source of ROS with levels 5–10 times higher than those in the cytosol. These ROS are generated from multiple sites within mitochondria, including ETC complexes, oxoglutarate dehydrogenase, and monoamine oxidase, among others. Under normal physiological conditions, mitochondria possess a robust antioxidant defense system that effectively counteracts mtROS. This antioxidant system comprises several enzymes and metabolites, such as superoxide dismutase, catalase, glutathione peroxidase, glutathione, ascorbate, and the thioredoxin system, working in concert to prevent mtROS accumulation [11,12]. Despite being associated with cellular damage when present in excess, ROS also have essential biological functions under normal conditions. They serve as secondary messengers in various signaling pathways within the cell, contributing to the regulation of crucial processes such as cell proliferation, differentiation, immunity, autophagy, and apoptosis. Moreover, ROS can modulate the activity of specific proteins and influence gene expression, acting as signaling molecules that ensure proper cell cycle regulation [13,14,15]. By precisely balancing the production and scavenging of ROS, cells can harness their signaling properties while preventing oxidative stress. This delicate equilibrium between ROS generation and the antioxidant defense system is vital for maintaining cellular homeostasis and ensuring the overall health and functionality of the cell.

The lipid composition of mitochondrial membranes is similar to that of other membranes, including the plasma membrane. They consist of phospholipids, such as phosphatidylcholine, phosphatidylethanolamine, phosphatidylinositol, phosphatidylserine, and phosphatidic acid. In addition, mitochondrial membranes also contain phosphatidylglycerol and cardiolipin, which are exclusively located in the mitochondria [16]. Alterations in the phospholipid composition can affect mitochondrial membrane integrity, permeability, and fluidity, hence, the stability and activity of membrane-associated proteins. Thus, the structure, dynamics, and function of mitochondrial membranes depend on the interaction of protein complexes with membrane lipids [16].

The development of signaling cascades involving mitochondrial membrane components is impacted by non-bilayer-forming phospholipids, such as cardiolipin [17]. As previously noted, cardiolipin is widely recognized as the distinctive phospholipid of mitochondrial membranes, with the two membranes displaying significant variations in cardiolipin concentration. Cardiolipin is found at a higher concentration in the IMM, constituting over 15% of the membrane phospholipids. In contrast, the proportion of cardiolipin in the OMM is estimated to be around 2–5% of the total lipids [18,19,20,21]. The unique structure of cardiolipin, a diphosphatidylglycerol combined with four acyl chains, results in a particular conical arrangement capable of mediating multiple interactions. Cardiolipin interaction with IMM proteins, enzymes, and carriers is essential for their stabilization and structural preservation [22,23,24]. Cardiolipin is actively involved in ETC integrity and activity, thus promoting the normal assembly and stabilization of mitochondrial ETC supercomplexes within the IMM [25,26]. Electrophoretic [27,28], kinetic [29], and structural [30] studies provide strong evidence that cardiolipin is critical to the structural organization, stability, and function of mitochondrial ETC (or respiratory) supercomplexes [31]. Additionally, cardiolipin helps promote mitochondrial health through mitochondrial fusion activity. The lipid has been demonstrated to promote the binding of optic atrophy 1 protein (OPA1) between distinct mitochondria [32].

Cristae structure and mitochondrial function are interconnected, as changes in cristae morphology can impact the stability of ETC complexes located within the IMM [33]. Mitochondria undergo internal structural reorganization by modifying cristae morphology [34] through a process known as “cristae remodeling”. The morphology of cristae can be characterized by the curvature of two distinct regions: the CJs, which are slender tubular structures that link the cristae to the inner boundary membrane, and the cristae lumen. Cristae remodeling involves a transition towards a membrane curvature (positively inclined), leading to the widening of the cristae. The morphological changes destabilize ETC complexes and reduce OXPHOS efficiency, proving that cristae shape is a critical morphological component controlling mitochondrial functions, such as respiration and cell death [35].

CJs prevent the release of cytochrome c from the inner membrane space, which impairs cell death signaling. However, the proapoptotic members of the Bcl-2 family can play a detrimental role in releasing cytochrome c by widening the CJs and inverting the curvature of cristae [36,37]. Therefore, variations in cristae structure affect mitochondrial bioenergetics. Indeed, cristae reshaping is associated with changes in the energetic state of the cell, mediating either cell survival or death signaling [38,39,40,41,42]. The regulation of mitochondrial cristae dynamics and formation is governed by OPA1, the mitochondrial contact site, and the cristae organizing system (MICOS) [43]. The following sections will focus on the role of OPA1 and MICOS in regulating the IMM structure.

#### 2.1.1. The Role of OPA1

The IMM possesses a significant abundance of proteins with a protein-to-lipid ratio of approximately 80:20. These proteins play crucial roles in various processes, including the ETC, OXPHOS, energy transfer, ion transport, and membrane structural organization. One of the central modulators of cristae curvature is the IMM dynamin-related protein OPA1. OPA1, a fusion GTPase protein, was named after its association with autosomal dominant atrophy, characterized by the degeneration of retinal ganglion cells and optic nerve, leading to gradual vision impairment. OPA1 mediates IMM fusion, a process in which mitochondrial matrix material is exchanged and a singular elongated mitochondrion is generated [44].

In addition to its role in IMM fusion, OPA1 is responsible for regulating cristae morphogenesis. Consequently, it has been recognized as important in regulating crucial mitochondrial processes, such as cellular apoptosis, respiratory efficiency, and genomic integrity. Thus, OPA1 plays a vital role in the maintenance and stability of the cristae structure under physiological conditions [38]. Cristae stability is essential for the regular operation of the ETC, and OXPHOS that supply the ATP required for the rhythmic contraction of myofibrils in cardiomyocytes. Due to alternative splicing and cleavage, OPA1 has eight isoforms ubiquitously expressed in humans. However, the amount of each isoform varies depending on tissue type. OPA1 processing involves the conversion of long-OPA1 (L-OPA1) to short-OPA1 (S-OPA1). In the heart, only five of the variants are present; two L-OPA1 isoforms are further cleaved into three soluble forms of S-OPA1 [8,45].

The different isoforms of OPA1 oligomerize and maintain tight CJs to prevent cytochrome c release [37], thus playing an important role in apoptotic-regulated cell death. OPA1 can be processed at two cleavage sites, S1 and S2, by two different peptidases in the IMM (Figure 2). Under physiological conditions, L-OPA1 is cleaved by the ATP-dependent metalloprotease YME1L to form S-OPA1. However, under stress conditions, YME1L is rapidly degraded, and the ATP-independent zinc metalloprotease OMA1 is activated and becomes the primary enzyme for OPA1 cleavage [46]. OPA1 is cleaved at different amino acid sequence sites. OMA1 cleaves OPA1 at the S1 in exon 5 site, whereas YME1L cleaves it at the S2 site in exon 5b [47]. While L-OPA1 has been shown to preserve cristae ultrastructure and the energy provision of mitochondria, its cleavage into S-OPA1 is associated with altered metabolic activity, secondary to mitochondrial fragmentation and energy depletion by IMM depolarization [48,49]. However, recent genetic studies have found S-OPA1 to maintain both cristae structure and respiratory activity through compensatory mechanisms, even when mitochondria lack fusion capacity [50,51,52,53]. The role of S-OPA1 in the structural organization of mitochondrial cristae and regulation of respiratory function remains unclear.

Regulating mitochondrial fusion and the formation of CJs is essential for the quality control of mitochondria and the viability of cells. One of the critical functions of OPA1 is the prevention of mitochondrial swelling by maintaining cristae structure and regulating the mPTP. However, the precise mechanism underlying these functions remains to be elucidated [54]. As previously mentioned, mitochondrial OPA1 processing occurs through the enzymatic activity of YME1L and OMA1. Under cellular stress, OMA1 activation has been demonstrated to cleave L-OPA1 isoforms leading to mitochondrial fragmentation, an underlying factor for the pathogenesis of many diseases such as cardiac IR injury [55,56]. However, YME1L cleavage of OPA1 under physiological conditions does not affect mitochondrial morphology. Genetic studies have demonstrated that the absence of YME1L in MEFs results in fragmented mitochondria that are unable to maintain cristae structure. Conversely, the depletion of OMA1 does not produce such effects [57]. Delayed onset of respiratory dysfunction has been reported in spinal cord mitochondria deficient in YME1L [58].

Interestingly, when fed a standard diet, OMA1-deficient mice did not demonstrate respiratory dysfunction in liver mitochondria [59]. This implies that the specific cleavage sites of OPA1 (S1 and S2 sites) may have varying functions concerning mitochondrial activity. As a result, YME1L and OMA1 may possess different functions that are not dependent on OPA1 processing. Loss of OPA1 in cultured cells has been shown to increase the susceptibility of mitochondria to cytochrome c release and apoptosis, whereas OPA1 overexpression is protective against multiple pathologies, including cardiac IR [51,60,61]. In mouse models with a 50% reduction in OPA1 protein levels, the mitochondrial network analyzed through electron and fluorescence microscopy appears notably affected, while cardiac function remained unaltered [54]. Despite extensive studies, the role of OPA1 function in the fusion and remodeling of cristae and mitochondria-mediated cell death in cardiac IR injury remains to be elucidated.

Cristae morphology profoundly affects mitochondrial respiratory capacity as it constitutes the IMM’s primary region that facilitates energy conversion [38,62]. OPA1 oligomers maintain a negative cristae curvature, and the stability of ETC complexes is critical for respiratory efficiency. Our most recent studies suggested the important role of both L-OPA1 and S-OPA1 in regulating mitochondrial respiration. Given the myriad effects of OPA1 on mitochondrial function, maintaining L-OPA1 integrity could prove beneficial under stress conditions such as Ca^2+^-induced mitochondrial swelling. We have previously shown that Ca^2+^-induced mitochondrial swelling stimulates proteolytic cleavage of L-OPA1 in isolated cardiac mitochondria [63]. Recently, Myls22, an OPA1 inhibitor, and TPEN (N, N, N’, N’-tetrakis(2-pyridylmethyl) ethylenediamine), a Zn^2+^-chelator and OMA1 inhibitor [64], exhibited protective effects against breast cancer growth [65] and attenuated mitochondrial damage [66]. Despite these promising effects, the role of OPA1 and OMA1 inhibition under Ca^2+^-induced mitochondrial swelling is yet to be evaluated. Furthermore, OPA1 oligomerization may be more critical for sustaining mitochondrial cristae structure than OPA1 cleavage. This was confirmed by the genetically modified MEFs, which preserved a moderate cristae structure with only S-OPA1 expression [50]. Moreover, experiments involving the starvation of mice have also highlighted the importance of OPA1 oligomerization as a crucial factor in preserving the structural integrity of mitochondria [41]. Additionally, OPA1 oligomerization is targeted during BID activation via apoptosis, contributing to the release of cytochrome c and debilitating the structural integrity of mitochondrial cristae [37,67]. Overall, the role of OPA1 oligomerization and its relationship to mitochondrial cristae structure and respiration must be further explored.

#### 2.1.2. The Role of MICOS Proteins

To maintain stability within mitochondrial CJs, mitochondria conserved a specific protein complex called MICOS [62]. This heterooligomeric protein complex is mainly found in CJs. MICOS controls the IMM architecture through direct membrane shaping, formation of contact sites, and biogenesis of proteins and lipids. When this multicomplex is disrupted, the cristae membrane loses connection to the inner boundary membrane, generating elongated structures that display aberrant architecture. These changes are associated with respiration defects. In mammals, seven subunits of the MICOS complex have been identified so far: Mic-60, Mic-10, Mic-19, Mic-25, Mic-26, Mic-27, and Mic-13 [62,63].

The MICOS protein complex comprises two subcomplexes associated with Mic-60 and Mic-10 subunits, which exhibit membrane-shaping capabilities. Subunit Mic-60 is known as mitofilin in humans and Fcj1 in yeast. Mic-60/mitofilin is an 80–90 kDa mitochondrial protein that is highly prevalent in human cells. It is anchored to the IMM through an amino-terminal transmembrane segment that allows interaction with Mic-19. Mic-60/mitofilin was initially identified as a mitochondria-linked “heart muscle protein”, as it is preferentially expressed in cardiac tissue [68,69]. Indeed, Mic-60 imposes curvature onto the IMM by an amphipathic alpha-helix that interacts with the IMS leaflet of the IMM. Also, Mic-60 is necessary for proper mitochondrial structure and function in many tissues, and its absence has been linked to mitochondrial malfunction, oxidative stress, and organ damage [68]. Deletion of Mic-60 results in disruption of cristae morphology as well as mitochondrial network fragmentation and activation of apoptosis through cytochrome c release and other pro-apoptotic proteins. It has been shown that Mic-60 interacts with the mitochondrial transcription factors TFAM and TFB2M, thus playing a role in mitochondrial DNA transcription [70]. Notably, the deficiency of Mic-60 results in a reduction of TFAM binding and recruitment of mitochondrial RNA polymerase to the promoters of mtDNA. Overexpression of Mic-60 impaired mitochondrial dynamics by promoting mitochondrial fusion and inhibiting mitochondrial fission, thus leading to increased mitochondrial respiration and ATP production in dopaminergic cells [71]. Furthermore, Mic-60 overexpression has been associated with a protective effect against rotenone-induced toxicity, which can occur due to reduced oxidative stress and mitochondrial dysfunction [71].

In addition to its role in membrane bending, Mic-60 also interacts with other MICOS subunits, such as Mic-19 and OMM proteins, such as the TOM complex, SAM, and the mitochondrial porin VDAC. It has been demonstrated that Mic-60 knockdown dramatically decreases the protein levels of other MICOS components, including members of the SAM complex. Genetic studies have shown that the Sam50-Mic19-Mic-60 complex is essential for cristae formation and maintenance through the interaction between OMM and IMM [72]. Furthermore, the downregulation of Mic-60 or Mic-19 leads to other MICOS components’ instability, the MICOS complex’s disintegration, and disordered mitochondrial cristae [72]. Thus, it appears that Mic-60 and Mic-19 interaction is essential for cristae stability in mammals. In this interaction, Mic-19 appears to regulate Mic-60 functions [73].

Mic-60 homeostasis is regulated by YME1L, a mitochondrial protease that also controls the turnover of OPA1. Indeed, the decrease in Mic-60 expression affects the protein levels of OPA1 [74]. Because OPA1 loss alters cristae structure, it has been suggested that OPA1 can form an oligomeric complex with MICOS-specific subunits, which disassemble when CJs go through remodeling and change shape [75]. Moreover, together with the F_O_F_1_-ATP synthase, OPA1 stabilizes tubular CJs and regulates the precise location of the MICOS complex and CJs [76]. Mic-60 levels are notably high at the CJs, whereas the levels of OPA1 are evenly distributed across the IMM. However, the expression levels of Mic-60 are increased in OPA1-deficient cells, leading to enhanced cellular resistance against the initiation of apoptosis [75].

Another core component of MICOS is the subunit Mic-10, which also induces membrane curvature due to its two transmembrane helices and interacts with other associated subunits to form a subcomplex within MICOS. The Mic-10 subcomplex comprises Mic-12, Mic-26, and Mic-27, which are crucial for generating stable CJs. The role of all independent components of MICOS was evaluated by selective expression of Mic-60 and Mic-10 genes using super-resolution light and 3D electron microscopy [76]. This study demonstrated that the Mic-60 subcomplex is necessary for CJ formation, whereas the Mic-10 subcomplex regulates the development of lamellar cristae. Interestingly, a study performing a Mic-60 knockdown showed mitochondrial swelling and cristae breakdown as well as an increase in ROS production and mitochondrial calpain activity. Also, Mic-60 knockdown cells exhibited a significant decrease in intracellular ATP production and ΔΨ_m_ [77] suggesting that Mic-60 plays a role in the pathological process of mitochondrial swelling. Nevertheless, further research is required to elucidate the precise function of Mic-60 in this process and to examine whether Mic-60 or other MICOS proteins would be feasible targets for reducing mitochondrial swelling caused by cardiac IR injury.

## 3. Ionic Regulators of Mitochondrial Matrix Volume

Under physiological conditions, ion transport (i.e., Na^+^, K^+^, Mg^2+^, H^+^, and Ca^2+^) through the IMM regulates mitochondrial matrix volume [78,79,80]. Slight changes in matrix volume promote mitochondrial activity and stimulate metabolism. Indeed, increases in matrix volume have been shown to stimulate fatty acid oxidation as well as ETC and OXPHOS. Moreover, mitochondrial swelling induces gluconeogenesis by stimulating pyruvate carboxylase [81,82]. Although the exact mechanisms for these effects of mitochondrial swelling are still being elucidated, IMM structural and functional remodeling could be involved in a response against oxidative stress. Hence, numerous oxidative stress-related human disorders, such as cardiovascular and neurological diseases, are influenced by mitochondrial swelling. Among ions, K^+^ and Ca^2+^ have been identified as the primary culprits of mitochondria matrix volume changes [79,83]. To keep in line with this, multiple influx and efflux mechanisms are involved in facilitating the transport of K^+^ and Ca^2+^ ions across the IMM (Figure 3). In the following sections, the K^+^ and Ca^2+^ transport mechanisms in mitochondria will be discussed. Main modulators of the mitochondrial matrix volume are shown in Table 1.

### 3.1. Transport Mechanisms for Mitochondrial K^+^

Mitochondrial physiological swelling is changes in matrix volume that help mediate mitochondrial function and structure. The regulation of mitochondrial matrix volume is a crucial process involving transporting K^+^ into and out of the matrix. Mitochondrial K^+^ transport is a complex and dynamic process that affects various cellular activities, such as bioenergetics, Ca^2+^ regulation, and ROS formation [84]. Therefore, K^+^ transport could be a target to reduce mitochondrial dysfunction after cardiac IR injury.

The identification of molecules that selectively target K^+^ channels and the lack of information on the significance of each channel to mitochondrial function constitute important obstacles in the development of mitochondrial K^+^ therapeutics. One K^+^ efflux and five K^+^ influx channels have been identified within the IMM of cardiac mitochondria (*reviewed in* [85]). The functional characteristics of these channels and their significance to the heart under physiological and pathological conditions are provided below.

#### 3.1.1. Mitochondrial ATP-Sensitive K^+^ Channels

The mitochondrial ATP-sensitive K^+^ (mK_ATP_) channels are a specific type of ATP-sensitive K^+^ channel of the IMM of cardiac cells. The mK_ATP_ is a molecular entity that comprises two distinct subunits, namely, the pore-forming subunit (MITOK) and the ATP-binding subunit (MITOSUR) [86]. The functionality of the mK_ATP_ channel is reliant upon various physiological factors, including but not limited to the ATP/ADP ratio, levels of GDP, Mg^2+^, PKC, and nitric oxide [87,88,89,90]. Several patch-clamp studies conducted on rat liver mitochondria indicate the involvement of this channel in regulating mitochondrial matrix volume. However, the activation of the mK_ATP_ channel has been shown to result in slight membrane depolarization that does not significantly prevent Ca^2+^ influx [91,92]. Despite the limited impact on Ca^2+^ uptake under physiological conditions, cardiac IRI studies reveal that activating mK_ATP_ channels effectively prevents Ca^2+^ accumulation in mitochondria [93,94]. It is thus necessary to re-evaluate the dependence of ΔΨ_m_ on K^+^ influx and to determine what effect, if any, ΔΨ_m_ has on Ca^2+^ influx. Alternatively, Ca^2+^ influx into the mitochondrial matrix could depend on the inhibition of MCU activity and modulation of mPTP [95]. These topics are open for investigation.

Notably, in cardiac mitochondria from rabbits, the activation of mK_ATP_ under conditions of anoxia/reoxygenation or substrate deprivation effectively inhibited the opening of mPTP by decreasing Ca^2+^ accumulation in the matrix [96]. Indeed, mK_ATP_ activation may play a role in cardioprotection via reducing Ca^2+^ overload and ROS production, as shown in IR experiments with rats treated with vitamin C [97]. In contrast, the mK_ATP_ channel opener (diazoxide) and blocker (5-hydroxydecanoate) have effects on mitochondrial matrix volume and respiration that are independent of their action on the mK_ATP_ channel during cardiac IR [98]. These results suggest additional mechanisms that affect matrix volume in the heart. It is also possible that mK_ATP_ channel inhibitors have non-specific effects on matrix volume. Nevertheless, it should be noted that vitamin C, an antioxidant, could also reduce ROS production by mK_ATP_ [99]. Furthermore, activating mK_ATP_ in rat cardiac mitochondria enhances respiration, increases ROS production due to matrix pH changes, and protects the heart against IR injury [100]. This study suggests that mK_ATP_ channels can be cardioprotective against IR injury despite their effect on ROS generation. Additionally, it has been demonstrated that the modulation of the AKT-Foxo1 signaling pathway by mK_ATP_ could be involved in ameliorating diabetic cardiomyopathy. Thus, in a murine model of type 2 diabetes (db/db mice), this pathway appears crucial in ameliorating cardiac function and preventing apoptosis [101]. Studies on liver mitochondria and HeLa cells in which genetic ablation of MITOK, an mKATP complex subunit, induced instability of ΔΨ_m_, widened the intracristal space, and decreased oxygen consumption rates [86]. Altogether, studies suggest that mK_ATP_ channels could be involved in various pathological processes. Further studies are required to establish the mechanisms by which mK_ATP_ channel activation, mPTP activation, ΔΨ_m_ loss, and ROS production crosstalk under physiological and pathological conditions such as cardiac IR injury.

#### 3.1.2. Mitochondrial Large-Conductance Ca^2+^-Activated K^+^ Channels

Mitochondrial large-conductance Ca^2+^-activated K^+^ (mBK_Ca_) channels are encoded by the Kcnma1 gene and are present in the IMM [102]. mBK_Ca_ channels, akin to their plasma membrane counterparts large-conductance Ca^2+^-activated K^+^ (BK_Ca_), exhibit activation in response to elevations of Ca^2+^ concentrations in the cytoplasm. Upon activation, mBK_Ca_ channels facilitate the influx of K^+^ into the mitochondrial matrix, thereby inducing reductions of the ΔΨ_m_ and impairing Ca^2+^ overload [103]. The modulation of specific isoforms of mBK_Ca_ activity in human glioma U-87 MG cells appears to be associated with cristae remodeling through alterations in membrane tension and shape [104].

**Table 1 antioxidants-12-01517-t001:** A list of compounds that target mitochondria to modulate matrix volume.

*Name*	*Mechanism of Action*	*Reference*
**mPTP inhibitors**		
Cyclosporin A	Inhibits CypD-ANT interaction and blocks mPTP opening	[105,106]
Sanglifehrin A	Inhibits CypD activity and CypD-ANT/PiC interaction, and blocks mPTP opening	[105,106,107]
N-methyl-4-isoleucine-cyclosporin (NIM811)	Inhibits CypD activity, and blocks mPTP opening; has no immunosuppressive activity	[108,109]
Bongkrekic acid	Inhibits ANT and prevents mPTP opening	[110]
Alisporivir (Debio-025)	Inhibits CypD activity, and blocks mPTP opening; has no immunosuppressive activity	[111]
**Mitochondrial Ca^2+^ channel inhibitors**		
Ru360 and Ru265	MCU inhibitors, prevent mitochondrial Ca^2+^ overload	[112,113]
**Mitochondrial K^+^ channel modulators**		
Diazoxide	mK_ATP_ opener, which increases K+ efflux from the matrix into the IMS leading to IMM hyperpolarization	[114,115]
Gibenclamide	mK_ATP_ opener channels, which leads to depolarization of the cells and insulin secretion	[116]
5-Hydroxydecanoate	mK_ATP_ inhibitor	[117]
NS11021 and NS1619	mBKCa activators	[118]
Paxilline and Iberiotoxin	mBKCa inhibitors	[119,120]
DCEBIO	An activator of intermediate conductance Ca^2+^-activated K^+^ channels (IKCa1/KCa3.1)	[121]
Retigabine and Flupirtine	Kv7.2–7.5 (KCNQ2–5) openers	[122,123]
XE991	Kv7 (KCNQ) channel blocker	[124]

Following its translocation to the mitochondria, the mBK_Ca_ is recognized by Tom22 located in the OMM and, subsequently, is imported to the IMM, where it interacts with the adenine nucleotide translocator (ANT) [125]. This suggests that mBK_Ca_ channels play a role in modulating energy generation and mPTP activation via physical interactions. In left ventricular cardiac mitochondria of rats, the functional modulation of mBK_Ca_ has been attributed to the BK-β1 subunit [126]. The study also revealed that the knockout of mBK_Ca_ renders the mitochondria more susceptible to Ca^2+^, thereby promoting the opening of the mPTP. According to a study on Kcnma1 knockout mice, activating the mBK_Ca_ channel using the BK_Ca_ agonist NS1619 induced cardioprotective effects against ischemic injury [127]. There is a consensus that mBK_Ca_ channels significantly regulate mitochondrial metabolism, ROS generation, and pathways leading to cell death. Indeed, the dysregulation of mBK_Ca_ channel activity has been associated with several pathologies, such as cardiac IR injury and cancer [128,129,130]. Nevertheless, the precise mechanisms through which mBK_Ca_ channels participate in these pathologies are not fully understood.

#### 3.1.3. Mitochondrial Small-Conductance Ca^2+^-Activated K^+^ Channels

Mitochondrial small-conductance Ca^2+^-activated K^+^ (mSK_Ca_) channels are also found in the IMM. In contrast to mBK_Ca_ channels, which facilitate the flow of currents with high conductance, mSK_Ca_ channels are responsible for the conduction of currents with low conductance. There are three isoforms of SK_Ca_ channels in the cell, namely, SK_Ca_1-3 [131]. The SK_Ca_3 isoform (mSK_Ca_) is localized in mitochondria, and its activation is triggered by elevations in the intracellular Ca^2+^ concentrations, resulting in K^+^ entry into the mitochondrial matrix. mSK_Ca_-mediated K^+^ uptake regulates mitochondrial volume, respiration, and ΔΨ_m_ [132]. The activation of mSK_Ca_ from guinea pig heart by DCEBIO induced a 35% increase of K^+^ influx compared to valinomycin, a K^+^ ionophore [133]. The data suggest that mSK_Ca_ may be important for the modulation of K^+^ distribution in the cell and for regulating mitochondrial matrix volume. mSK_Ca_ channels could also be significantly involved in regulating mitochondrial Ca^2+^ homeostasis and ROS production, and its dysregulation has been suggested in pathologies such as heart failure [134,135]. In Langendorff-perfused guinea pig hearts, DCEBIO decreased mitochondrial Ca^2+^-uptake and enhanced cardiac function following IR injury [133]. Likewise, DCEBIO reduced infarct size, while the mSK_Ca_ inhibitor NS8593 worsened heart function in rats with an in vivo left anterior descending artery occlusion [131]. Understanding the specific mechanisms through which mSK_Ca_ channels participate in developing these diseases remains elusive and requires further research.

#### 3.1.4. Mitochondrial Voltage-Gated K^+^ Channels

The voltage-gated K^+^ channels encoded by the Kv7 gene (Kv7) family comprises five distinct channels, namely, Kv7.1 through Kv7.5. While Kv7.1 functions as a channel on the cardiac cell membrane, Kv7.2 through Kv7.5 are primarily expressed in neurons [136]. In addition, it should be noted that the mitochondrial voltage-regulated K^+^ channels, specifically Kv7.4, are located in the IMM. However, it is important to acknowledge that these channels are not ubiquitous across mitochondrial populations or tissue types. For instance, cardiac and neuronal mitochondria contain mKv7.4, but liver mitochondria do not. Similar findings have been reported in previous studies [137,138]. Indeed, approximately 30–40% of mitochondria in H9c2 cardiomyoblasts were found to contain mKv7.4 [138]. The mKv7.4 channel is relevant for regulating the electrical potential across the mitochondrial membrane and, hence, could impact various cellular processes such as ATP production, Ca^2+^ signaling, and ROS production [137]. Currently, limited knowledge exists regarding the function of mKv7.4 in cardiac tissue. However, retigabine, a Kv7 channel activator, enhanced cell viability and cardiac recovery from ischemic damage in both H9c2 cells and male rat hearts. [138]. Further investigation is necessary to assess the precise involvement of mKv7.4 in IR injury, given that both Kv7.1 and mKv7.4 are present in cardiac tissues and are influenced by retigabine. Thus, ongoing investigations about Kv7.4 channels have generated promising outcomes, indicating their potential as a therapeutic target in various medical conditions.

#### 3.1.5. Mitochondrial Na^+^-Activated K^+^ Channels

Mitochondrial Na^+^-activated K^+^ (mSlo2) channels, a novel type of ion channel, have been identified in the IMM. The activation of these channels is triggered by elevated levels of intracellular Na^+^, which facilitates the influx of K^+^ to the mitochondrial matrix. The presence of the mSlo2 channel in cardiac mitochondria was supported by Western blot [139]. A study revealed that cardiac mSlo2 exhibits a conductance of approximately 138 pS [140]. Nonetheless, no cardioprotective effects of mSlo2 were observed in that study following a challenge with isoproterenol in a Langendorff model [140]. This observation suggests that the involvement of mSlo2 in safeguarding the heart against mitochondrial swelling may not be substantial. However, another study utilizing Bithionol revealed that activation of mSlo2 channels results in decreased infarct size in mice hearts after IR injury. [139]. Therefore, there is insufficient data on the role of mSlo2 in cardiac function. For these reasons, whether mSlo2 can be considered a viable therapeutic target for IR injury is an open question.

#### 3.1.6. K^+^/H^+^ Exchanger

The mitochondrial K^+^/H^+^ exchanger (KHE) is an IMM protein that plays a critical role in regulating the transport of K^+^ and could be critical for regulating mitochondrial volume and metabolism [141]. The KHE is inhibited by Mg^2+^, quinine, and ETC inhibitors and is modulated by changes in osmolarity, pH, and ΔΨ_m_ [142,143,144]. The Wolf–Hirschhorn syndrome is a medical condition characterized by a genetic mutation in the LETM1 gene, leading to the production of KHE. This syndrome has been found to exhibit mitochondrial deficiencies, including a reduced ΔΨ_m_, altered Ca^2+^ homeostasis, and decreased ATP levels [145]. Considerable debate exists regarding the predominant coding of KHE, given that LETM1 also appears to encode the mitochondrial Ca^2+^/H^+^ exchanger (mCHE). However, the recent finding that TMBIM5, a protein found in mitochondria, exhibits mCHE properties could help resolve the controversy regarding the LETM1 coding of KHE [146]. At present, information on the impact of KHE in cardiac IR injury is lacking. The presence of cardiac anomalies, including atrial and ventricular septal defects and hypoplastic left heart syndrome, in individuals with Wolf–Hirschhorn syndrome highlights the relevance of studying KHE’s involvement in cardiac development and injury [147].

Therefore, K^+^ channels comprising one efflux and five influx channels located in the IMM of cardiac mitochondria appear relevant in both physiological and pathological states. However, which of these channels is the predominant form that governs cardiac function remains to be elucidated. The latter is aggravated by the limited availability of pharmacological agents that selectively target mitochondrial K^+^ channels. Therefore, future studies should focus on developing drugs to target these channels and investigating the role of mitochondrial K^+^ channels (i.e., KHE, mSlO2, and mKv7.4) in regulating cardiac function.

### 3.2. Mitochondrial Ca^2+^ Transport Mechanisms

#### 3.2.1. Mitochondrial Ca^2+^ Influx Mechanisms

The cardiac muscle mitochondria harbor three distinct pathways for the influx of Ca^2+^, namely, the mitochondrial Ca^2+^ uniporter (MCU), rapid mode of Ca^2+^ uptake (RaM), and mitochondrial ryanodine receptor (mRyR). Several genetic and pharmacological studies [148,149,150] have established that the MCU is the primary channel for Ca^2+^ influx in cardiac mitochondria. The molecular constitution of the MCU complex described in 2011 comprises three key components, namely, the MCU, mitochondrial Ca^2+^ uptake 1–2 (MICU1-2), and essential MCU regulators (EMRE) [151,152,153]. The uptake of Ca^2+^ through the MCU is significantly dependent on the ΔΨ_m_ and is inhibited by Mg^2+^. The MICU subunits play a role in gatekeeping the MCU complex, regulating Ca^2+^ entry to prevent mitochondrial Ca^2+^ overload [154,155]. Studies performed with HEK293T cells recently found a direct interaction between the MCU complex and ETC complex I that helps mediate energy levels when complex I is impaired [156]. Besides regulating bioenergetic homeostasis through complex I, the MCU modulates mitochondrial volume through matrix Ca^2+^ accumulation. Indeed, impaired matrix Ca^2+^ accumulation was observed in cardiac mitochondria of MCU knockout mice that could not demonstrate mPTP opening [157]. Furthermore, this study also showed changes in Ca^2+^-induced mitochondrial swelling in liver, brain, and heart mitochondria that occurred only in the presence of the MCU, further confirming the critical role of MCU in mitochondrial volume regulation and that alternative Ca^2+^ influx channels only play a small role in regulating this parameter.

Pharmacological interventions targeting the MCU have produced favorable cardioprotective outcomes after IR injury. Indeed, the administration of Ru360, a selective inhibitor of the MCU, to rats subjected to an in vivo coronary artery ligation had a protective effect on cardiac and mitochondrial functionality [158]. A similar study conducted on mice involving coronary artery ligation [159] provides additional evidence that the administration of Ru360 reduces the extent of myocardial infarction and preserves the integrity and functionality of mitochondria following IR injury. Currently, the most significant challenges for available MCU-targeting products are their permeability in the cellular membrane and pharmacological specificity [160]. As a result, no clinical studies for these ruthenium compounds have been conducted.

#### 3.2.2. Mechanisms for Mitochondrial Ca^2+^ Efflux

Mitochondria play a critical role in regulating intracellular Ca^2+^ levels through several transport systems. One of them is the mitochondrial Na^+^/Ca^2+^ exchanger (mNCE, also denoted as NCLX or mNCX), an electrogenic transporter that exchanges three cytosolic Na^+^ for one mitochondrial Ca^2+^ [112,161]. The mNCE is an important mediator of Ca^2+^ signaling in cardiac cells, and its dysfunction has been implicated in the pathogenesis of IR injury. The extrusion of mitochondrial Ca^2+^ during physiological stimulations is limited by mNCE levels, while LETM1 levels are deemed insignificant [162]. For cardiac mitochondria, the mNCE, rather than the mCHE, may be the primary route for mitochondrial Ca^2+^ efflux [163]. The mNCE protein is a crucial cellular component, and its study offers a valuable understanding of the intricate connections between Ca^2+^ and redox signaling mechanisms [164]. Upregulation of mNCE provides protection against myocardial IR injury and ischemic heart failure [165]. The same study also demonstrated that the abolition of mNCE leads to left ventricular remodeling, heart failure, and death. These findings suggest that mNCE is essential for normal cardiac function. Although the targeting (activating) of mitochondrial Ca^2+^ efflux channels for IR injury shows therapeutic potential, there is a lack of pharmacological agents available for this purpose.

Therefore, studies indicate that preventing excessive Ca^2+^ uptake into the mitochondria via MCU inhibition contributes to preserving mitochondrial function and cellular viability. Mitochondrial targeting Ca^2+^ channel blockers such as ruthenium red products (Ru360 and Ru265) are promising for maintaining cardiac mitochondrial function during Ca^2+^ overload conditions [166,167]. However, MCU-targeting products are limited due to their membrane permeability properties and pharmacological specificity [160]. In contrast, increasing mitochondrial Ca^2+^ efflux could be beneficial for preventing mitochondrial injury in pathological conditions [165]. In this line of thought, developing drugs to increase mNCE activity could be promising for treating pathological conditions associated with mitochondrial Ca^2+^ overload.

## 4. mPTP: A Non-Selective Channel Involved in Mitochondrial Swelling

The mitochondria play a central role in mediating cell death through several pathways, such as apoptosis, ferroptosis, and mPTP-mediated necrosis [168,169]. The occurrence of mPTP-mediated necrosis is attributed to a significant disruption in mitochondrial function in response to energy or oxidative stress. This disruption involves insufficient ATP production, excessive ROS generation, and Ca^2+^ overload. These conditions attributed to the activation of mPTP lead to cell death. Various factors such as P_i_, ADP/ATP, pH fluctuations, Mg^2+^, and the activation of CypD and Bax/Bak potentially modulate the activity of mPTP [170,171]. The mPTP has been shown to work in both low-conductance (300 pS, reversible) and high-conductance (1.3 nS, irreversible) modes, both of which are reported to contribute to the dissipation of ΔΨ_m_ [172]. Also, the opening of mPTP results in substantial swelling, leading to OMM rupture and an increase in ROS production [173]. All factors that contribute to mPTP opening, including mitochondrial Ca^2+^ overload, ROS accumulation, ATP depletion, high Pi, and ΔΨm loss are present at reperfusion after cardiac ischemia. Therefore, mPTP opening occurs in the reperfused heart and plays a central role in the pathogenesis of cardiac IR (*reviewed in* [174,175]).

Numerous experimental and computational approaches have been employed to study the kinetics of mPTP activation and the shift from a low- to high-conductance state [176,177,178,179]. The utilization of these experimental approaches offers valuable insights into ion homeostasis, facilitating our understanding of the fundamental mechanisms contributing to mitochondrial dysfunction through swelling. Nevertheless, these methodologies have been unsuccessful in identifying the molecular composition of the mPTP. Over the years, there has been a growing interest in investigating mitochondria under both physiological and various pathological conditions (Figure 4A). However, it is noteworthy that the number of studies focusing on the mPTP has shown a declining trend since 2015 (Figure 4B). This decrease in research activity can be attributed to the longstanding challenge of identifying the precise molecular identity and regulatory pathways of the mPTP, which has remained elusive despite extensive research efforts since the 1980s. This lack of clarity has led to a waning interest in this area of study. Recent studies have suggested the possibility of different mPTP forms being formed by ANT or F_O_F_1_-ATP synthase, and a potential interaction between these proteins may contribute to their regulatory roles for each other. However, these findings require further investigation and validation. The limited understanding of the molecular structure of the mPTP has hindered the development of new pharmacological compounds targeting this pore. As such, there is a pressing need for additional research to unravel the complexities of the mPTP and identify potential therapeutic targets for drug development. This section discusses the status of current knowledge on the identity of the mPTP and its function in cardiac IR injury.

Notably, despite the lack of knowledge regarding the molecular identity of the mPTP, there is a consensus among scholars that CypD serves as an important protein regulator of the mitochondrial pore [180,181,182]. Various treatments have been developed to prevent cardiac IR damage by targeting CypD. These include cyclosporin A (CsA) and sanglifehrin A. [183,184]. CsA binds to CypD, a key regulator of mPTP, and prevents its interaction with other mPTP components [185]. By inhibiting mPTP opening, CsA helps maintain mitochondrial integrity, preserves ATP production, and reduces cell death. Like CsA, sanglifehrin A is an immunosuppressant that prevents mPTP-mediated mitochondrial swelling and effectively preserves mitochondrial function. However, unlike CsA, sanglifehrin A prevents CypD from interacting with ANT and phosphate carriers while blocking its PPIase activity [105]. Additionally, sanglifehrin A exhibits greater selectivity and a higher affinity for CypD, making it a potentially superior therapeutic agent for targeting mPTP opening [105].

Initially, several studies reported cardioprotective effects of CsA on rodent and rabbit models [108,186,187]. A meta-analysis of twenty in vivo experimental trials in reperfused myocardial infarction animal models (four species) found that CsA reduced infarct size, despite substantial variability in effect across studies. However, CsA did not affect infarct size in pig hearts raising concerns regarding the possible cardioprotective benefits of CsA in humans [188]. In support of this contention, a clinical study from 2011 to 2014 indicated that intravenous CsA did not improve clinical outcomes in patients with anterior ST-segment elevation myocardial infarction (STEMI) referred for primary percutaneous coronary intervention at one year of treatment, nor did it prevent adverse left ventricular remodeling [189]. Overall, owing to the failure of clinical studies to support a cardioprotective effect and to provide data that shows targeting of CypD, there are doubts on whether CypD is the optimum target for creating novel drugs (*reviewed in* [180]).

Recent studies suggest that a Ca^2+^-induced CsA-sensitive membrane depolarization independent of permeabilization may be linked to the low-conductance mode of mPTP [190,191]. The component responsible for this depolarization does not appear to be part of the molecular structure of mPTP but would be a gating precursor to the high-conductance mPTP, allowing it to open. This idea seems reasonable since CypD knockout mitochondria can still open mPTP, although at greater Ca^2+^ concentrations [192,193]. CypD could interact with other proteins to generate the low-conductance mPTP, which is independent of the high-conductance state. However, evidence is lacking in this regard.

The F_1_F_0_-ATP synthase [194,195], the phosphate carrier [196], and the ANT [193,197] have been suggested as probable components of the mPTP. Notably, all have been reported to interact with CypD. However, genetic ablation of the phosphate carrier indicates that it is a non-essential component of the mPTP [198,199]. Currently, the mPTP is hypothesized to be either two (or more) separate pores containing different proteins (a multiple pore hypothesis) or a complex of numerous proteins that produce the high-conductance pore (Figure 5) [200,201,202]. Most of these investigations have focused on two proteins, ANT and FOF1-ATP synthase (F-ATP synthase), that regulate two pores, A-MPTP and F-mPTP, respectively [202,203]. Although both pores appear necessary for mPTP function, it is critical to determine whether ANT and F_O_F_1_-ATP synthase operate in concert to produce the functional mPTP.

### 4.1. Adenine Nucleotide Translocase

In mitochondria, ANT has four isoforms expressed variably in various tissue types; however, only isoforms 1, 2, and 3 have been identified in cardiac mitochondria through Western blot [204,205]. In physiological conditions, ANT catalyzes the 1:1 exchange of newly synthesized ATP for cytosolically localized ADP across the IMM. However, it seems crucial in mediating mPTP opening in pathological conditions. Initially, it was shown that CypD and ANT interacted, which supported its role in the formation of the mPTP [206,207]. Then, this was validated by genetic deletion of ANT isoforms 1, 2, and 4 in mouse liver mitochondria, which provided protection against Ca^2+^ levels >1 mM. However, the opening of mPTP induced by Ca^2+^ occur at exceedingly high concentrations of Ca^2+^, whereas complete protection was solely attained when ANT deletion was combined with *Ppif* ablation. [193]. Furthermore, it is well known that research utilizing various ANT inhibitors revealed that the exchanger performs a divergent (opposite) function in mPTP development. Thus, bongkrekic acid, an ANT inhibitor that generates an m-state conformation, inhibits mPTP opening and protects against Ca^2+^-induced mitochondrial swelling [208]. Atractyloside, another ANT inhibitor that induces c-state conformation of the exchanger, was shown to sensitize mitochondria to mPTP opening [209]. In agreement with these studies, our laboratory conducted in vitro and in silico tests in our lab to explore the probable involvement of ANT in mPTP gating. The ANT model was included in a basic but successful empirical model of mitochondrial bioenergetics to determine the point at which Ca^2+^ overload triggers mPTP opening through an ANT switch-like mechanism triggered by matrix Ca^2+^ and blocked by extra-mitochondrial ADP [210]. Although the model demonstrates that ANT can act as a gating mechanism for the mPTP, it does not prove that ANT is the mPTP. Nevertheless, studies show that cardiac ANT can undergo a Ca^2+^-selective transformation into an unselective channel with a conductance of up to 600 pS [211,212,213]. However, patch-clamp experiments on liver mitoplasts have revealed conductance of up to 1.3 nS, roughly twice the magnitude observed on ANT [214].

The presence of conflicting evidence on this topic poses the following questions: Does the ANT protein operate as a low-conductance pore, or does it collaborate with other proteins in the IMM to create a high-conductance pore? Also, considering the distinct ANT isoforms, do individual ANT isoforms undergo a conversion into a non-selective channel, and if this is true, do they exhibit equivalent conductance? Ultimately, why is there complete prevention of low- and high-conductance mPTP formation when all ANT isoforms and CypD are inhibited? Hence, further studies are required to elucidate whether ANT could operate with other IMM proteins to activate the mPTP or if it works as a non-selective individual pore in physiological and pathological conditions.

### 4.2. F_1_F_O_-ATPase

F_1_F_O_-ATPase, also known as F-ATP synthase, is a multiunit enzyme complex that synthesizes ATP from ADP and P_i_ under aerobic conditions, using a proton-motive force generated by respiration. F_O_F_1_-ATP synthase is composed of the catalytic core, the F_1_ region, which contains five subunits (α, β, γ, δ, and ε), and the membrane-embedded F_O_ region that has at least three subunits (A, B, and C). Under physiological conditions, the F_O_F_1_-ATP synthase exists as a monomer, dimer, and oligomer. The F_O_F_1_-ATP synthase dimerization is shown to play a role in the formation and stability of the IMM cristae invaginations [215]. F_O_F_1_-ATP synthase dimers have been demonstrated to generate pores with conductance ranging from 1.0 to 1.3 nS, comparable to high-conductance mPTP [216]. However, the molecular mechanism of how the mPTP forms through the F_O_F_1_-ATP synthase dimer has not been defined. Several studies have attempted to narrow down specific F_O_F_1_-ATP synthase subunits that may be involved in mPTP formation. Knockout studies of the c subunit of F_O_F_1_-ATP synthase support a regulatory role of this subunit in mPTP opening [195]. Moreover, other studies establish the c subunit as a core component of the mPTP [217,218,219]. Furthermore, studies in HeLa cells evaluating the f subunit by downregulation suggest a role in modulating the size and Ca^2+^ sensitivity of the PTP. These findings indicate that subunit f may play a role in mediating mPTP opening [220]. T163S mutations in HeLa cells inhibited Ca^2+^-ATPase activity and prevented mPTP-dependent cell death, raising the possibility that the β subunit is also involved in activating the mPTP [221]. However, contrary to popular perception, multiple investigations targeting essential components of the F_O_F_1_-ATP synthase have shown that mPTP may still develop without these components [222,223,224]. Consequently, it is not likely that the F_O_F_1_-ATP synthase function as mPTP. However, the potential for F_O_F_1_-ATP synthase and ANT to collaborate in creating a channel within the ATP synthasome is a subject open for investigation.

The ATP synthasome is a multicomplex unit composed of F_O_F_1_-ATP synthase, ANT, and phosphate carrier at a stoichiometry of 1:1:1 [225]. The ATP synthasome are intended to promote ATP synthesis by bringing together the necessary components for this process. The ATP synthasome assembly modifications have been found to need CypD. For example, the study found that mPTP causes the ATP synthasome to disassemble in wild-type cardiac mitochondria. In contrast, cardiac mitochondria from CypD knockout mice showed the synthasome levels to be more resistant to disassembly [226]. These observations imply that CypD may be needed to facilitate changes in ATP synthasome conformation. Currently, only a few studies have focused on the potential role of the ATP synthasome as the high-conductance mPTP. However, demonstrating that ANT alone might be the high-conductance mPTP is difficult, given the current state of knowledge. This is due to c subunit knockout studies showing a low conductance state with similar properties as the ANT pore [191]. Furthermore, a recent patch-clamp study on proteoliposomes containing reconstituted submitochondrial vesicles that carry both ANT and ATP synthase has identified a channel that exhibits similarities to mPTP. The channel’s conductance decreased upon the inhibition of ATP synthase or ANT, indicating a cross-inhibition between the pores of ATP synthase and ANT [227]. Further research is necessary to determine the alleged function of ATP synthasome in developing mPTP in cardiac cells.

## 5. The Role of Cytoskeletal Proteins in Mitochondrial Swelling

The sarcomere, the fundamental contractile unit of cardiomyocytes, is intricately connected to both the nucleus and mitochondria. Within the cytoskeleton, actin filaments and microtubules play essential roles in interacting with mitochondria and regulating their positioning, movement, and fission–fusion events. While the exact mechanisms of crosstalk between cytoskeletal proteins and mitochondria are still under investigation (*reviewed in* [210,211,212,213]), it is evident that cytoskeletal proteins, such as actin and tubulin, may significantly contribute to the regulation of mitochondrial morphology, dynamics, and swelling (*reviewed in* [228,229,230,231]). Recent studies have indicated that actin filaments and associated proteins can influence mitochondrial swelling. Actin polymerization enhances ER-mitochondrial interaction, leading to increased mitochondrial Ca^2+^ levels through more efficient Ca^2+^ flow from the ER to mitochondria [232] potentially triggering mitochondrial swelling. Moreover, disrupting F-actin has been found to reduce DRP1 translocation to mitochondria, attenuating fission and indicating a close relationship between the actin cytoskeleton and mitochondrial fission [233]. Actin and myosin II further stimulate the recruitment of Drp1 by the OMM, where it oligomerizes and participates in mitochondrial fission. Additionally, the β-tubulin isotype II has been observed to colocalize with proteins such as VDAC, ANT, and mitochondrial creatine kinase (MtCK) [234,235], suggesting its potential role in maintaining mitochondrial physiology, including the regulation of mPTP and matrix volume changes via ANT, a primary regulator of mPTP. Similarly, actin filaments may also interact with proteins associated with mPTP, a key regulator of mitochondrial swelling. Microtubules serve as tracks for mitochondrial movement within the cell, helping to maintain their distribution. Disruption of microtubules can lead to altered mitochondrial dynamics and swelling [236]. Thus, cytoskeletal proteins indirectly play a role in regulating changes in the matrix volume of mitochondria due to their relationship with mitochondrial dynamics and shape. Nonetheless, the precise roles of cytoskeletal proteins in regulating the matrix volume of mitochondria in response to physiological and pathological stimuli necessitate further research for a comprehensive understanding. Continued investigation in this area promises to shed light on the intricate interplay between the cytoskeleton and mitochondrial function, which may hold valuable insights into cardiac health and disease.

## 6. Conclusions

In conclusion, the field of mitochondrial research has made significant progress in understanding the mechanisms underlying mitochondrial structural remodeling and swelling and its implications for various human diseases, including coronary heart diseases. However, further research is needed to understand how proteins crucial for preserving mitochondrial structural integrity get altered under pathological situations. Moreover, the molecular identity and regulation of mitochondrial ion channels, particularly K^+^ and Ca^2+^ channels, as well as the mPTP, are still undefined and, hence, subjects of ongoing investigation. The failure of clinical trials to demonstrate the protective effects of pharmacological agents targeting mPTP suggests that the therapeutic approach in future pre-clinical studies should emphasize targeting mitochondrial structural remodeling and swelling. However, this could be challenging considering the current limitations of drugs (mitochondrial membrane permeability and pharmacological specificity properties) developed to study these processes. Nevertheless, the advancement in understanding the molecular mechanisms of mitochondrial structural reorganization, ion transports, and mPTP induction will contribute to developing novel therapeutic strategies to maintain or improve cardiac function under stress and pathological conditions.

## Figures and Tables

**Figure 1 antioxidants-12-01517-f001:**
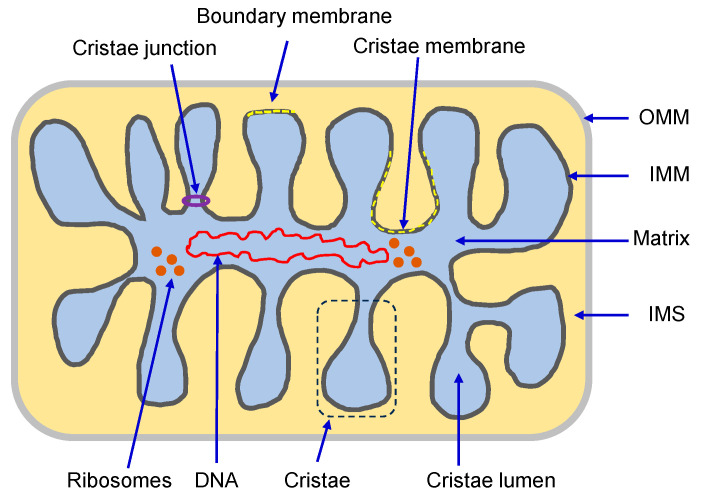
Structure of mitochondria (*see text for details*).

**Figure 2 antioxidants-12-01517-f002:**
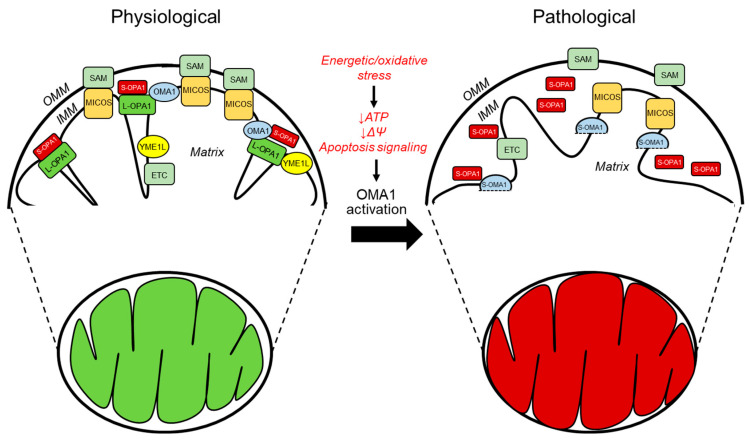
The primary regulators of mitochondrial cristae organization in both physiological and pathological conditions. The diagram depicts the structural arrangement of mitochondrial membranes, in which the cristae arise from the inward folding of the inner membrane towards the matrix. The MICOS complex, which is responsible for organizing mitochondrial contact sites and cristae, is situated at the CJs. It comprises seven subunits, namely, MIC10, MIC13, MIC19, MIC25, MIC26, MIC27, and MIC60. The stabilization of the CJs and establishing contacts between the IMM and OMM requires MICOS to be involved. The convergence of MICOS and the SAM complex results in the formation of a more extensive entity known as the mitochondrial intermembrane space bridging complex, which encompasses the intermembrane space. The protein OPA1 is highly localized at the CJs and its presence is crucial for preserving the proper dimensions of these structures. This necessitates the interplay between the membrane-bound long (L-) variants and the soluble short (S-) variants of OPA1. Under physiological (denoted in green) and pathological (denoted in red) conditions, the IMM proteases YME1L and OMA1 regulate OPA1. The protein OMA1 is responsible for cleaving all isoforms of OPA1 at the S1 site, whereas YME1L is responsible for cleaving OPA1 at the S2 site but only for the subset of OPA1 that contains the splice variant. Under physiological conditions, YME1L modifies the L-OPA1 to S-OPA1 ratio of the heterooligomeric complexes by cleaving a subset of the L-OPA1 isoforms at the S2 cleavage site. Under normal circumstances, OMA1 exhibits minimal activity; however, it is known to become activated in response to pathological conditions. OMA1 cleaves all L-OPA1 isoforms at the S1 cleavage site, releasing S-OPA1 from the IMM and OMA1 self-cleavage resulting in S-OMA1. Additionally, under stress conditions, the SAM-Mic19-Mic-60 axis can be disrupted through the cleavage of Mic19 by OMA1, resulting in the separation of the SAM and MICOS. Overall, the activation of OMA1 under pathological circumstances leads to impaired mitochondrial function.

**Figure 3 antioxidants-12-01517-f003:**
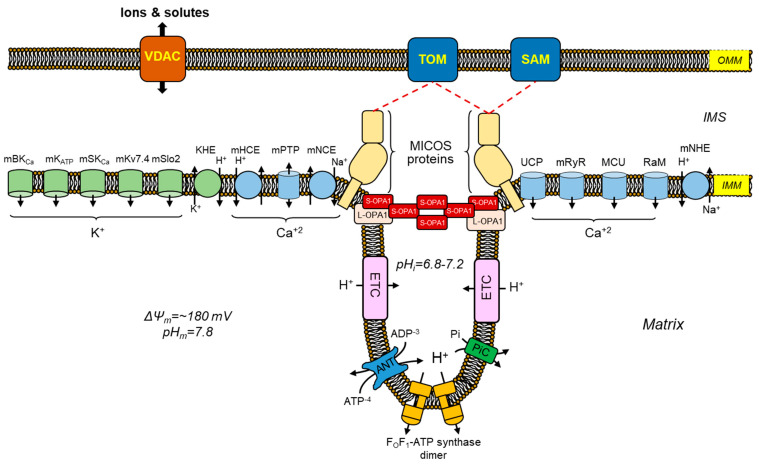
Mechanisms involved in maintaining ion homeostasis, structural maintenance, and matrix volume within the mitochondria. A graphical representation that illustrates the primary influx and efflux channels of Ca^2+^ and K^+^, which are responsible for regulating the volume, as well as the structural constituents that govern the morphology of cardiac mitochondria. The primary mechanisms responsible for Ca^2+^ influx are the MCU, RaM, mRyR, and uncoupling proteins 2 and 3 (UCP2/3). The maintenance of Ca^2+^ and ion homeostasis in the matrix is a crucial aspect, wherein the Ca^2+^ efflux mechanisms such as mitochondrial mNCE, mHCE, and mPTP play a significant role. The transportation of K^+^ holds equal significance for the metabolism and functioning of mitochondria. Alterations in the concentration of K^+^ exhibit a direct correlation with variations in the volume of the mitochondrial matrix. The mechanisms responsible for K^+^ influx comprise the mBK_Ca_, mK_ATP_, mSK_Ca_, mKv7.4, and mSlo2. Conversely, the K^+^ efflux mechanisms are restricted to the KHE. The function of OPA1 is the fusion of IMM and its regulatory role in cristae morphogenesis. In the heart, five isoforms of OPA1 exist that can be characterized as either L-OPA1 to S-OPA1. The various isoforms of OPA1 have demonstrated the capacity to oligomerize and uphold stringent CJs. A protein complex known as MICOS facilitates preserving a stable state in mitochondrial cristae junctions. In addition to its membrane bending capabilities, MICOS exhibits interactions with proteins situated in the OMM, including the translocase of the outer membrane (TOM) complex, the sorting and assembly machinery (SAM), and the voltage-dependent anion channel (VDAC) protein. Furthermore, the mitochondrial Na^+^/H^+^ exchanger (mNHE) has been implicated in the maintenance of ion homeostasis within mitochondria. Also, the invaginations of CJs are generated by dimeric F_O_F_1_-ATP synthase, where CJ structures serve as the site for ETC complexes, ANT, and phosphate carrier (PiC), which utilize the pH gradient to facilitate the process of ATP synthesis.

**Figure 4 antioxidants-12-01517-f004:**
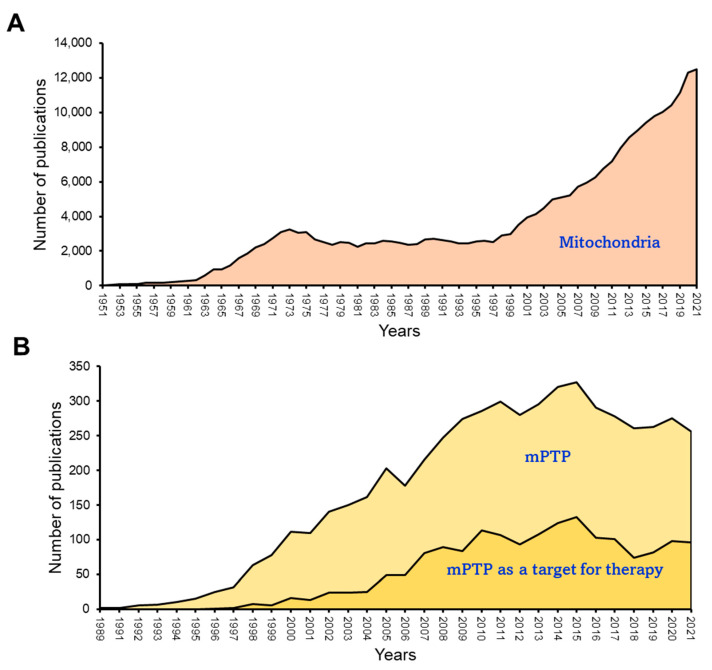
Publication history related to mitochondria and mPTP research. (**A**) Mitochondrial research conducted over time, indicating a steady increase in the number of studies. (**B**) A decline of approximately eight years in mPTP research and mPTP-derived therapeutic interventions. The data was obtained from PubMed (https://www.ncbi.nlm.nih.gov (accessed on 5 June 2023)).

**Figure 5 antioxidants-12-01517-f005:**
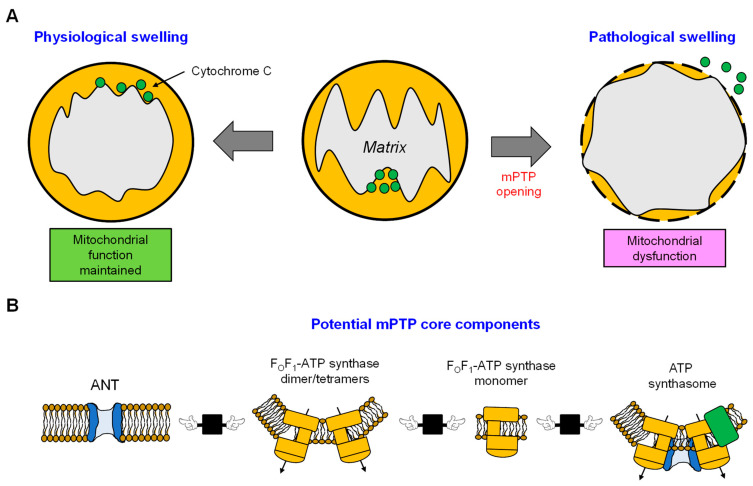
mPTP-mediated matrix swelling and potential mPTP models. (**A**) The opening of the mitochondrial permeability transition pore (mPTP) triggers mitochondrial swelling, leading to various consequences for mitochondrial function and cell survival, depending on the degree of swelling. (**B**) Although the molecular identification of the mPTP remains unclear, it is widely accepted that CypD acts as a protein regulator of the mPTP. Most studies have focused on a pair of proteins known as ANT and F_O_F_1_-ATP synthase (F-ATP synthase). The formation of the mPTP may be attributed to the synergistic or independent functioning of ANT and F_O_F_1_-ATP synthase. However, the participation of both entities is essential. It can be proposed that mPTP may be formed by one of the proteins or complexes, namely, ANT, F_O_F_1_-ATP synthase, F_O_F_1_-ATP synthase dimer/tetramer, or the ATP synthasome.

## Data Availability

Not applicable.

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
