# Peer review of "Mitochondrial Volume Regulation and Swelling Mechanisms in Cardiomyocytes"

_antioxidants, 2023, doi:10.3390/antiox12081517_

Round 1

Reviewer 1 Report

In this manuscript, Chapa-Dubocq et al. review the role of mitochondrial matrix volume in IMM remodeling and the crosstalk between these processes.

The review is informative and well written.

Comments:

1.      In the Motochondrial membrane and structure organization, a schematic figure illustrating mitochondrial membranes and compartments will help the reader to better understand mitochondria architecture.

2.      A table with information about pharmacological agents (and mechanism of action) modulating mitochondrial matrix volume would be also helpful.

3.      The role of cytoskeleton proteins in mitochondrial swelling would be an interesting point to address.

Reviewer 2 Report

In this review, Chapa-Dubocq et al. give an extensive overview not only over mitochondrial functions and structures in general, but also over celluar causes and consequences of mitochondrial swelling. They focus on cardiomyocyte mitochondria, but most parts of the review are very general and therefore useful also for a broader spectrum of readers. In general, this review is very exhaustive and maybe one of the best reviews I have red so far in “Antioxidants”.

Overall the writing, phrasing and grammar of the manuscript are excellent and understandable. Very well done. Nearly all of the topics are very detailed and sufficiently explained. I have only two minor suggestions to further improve this already great article.
